# Drought Drives Growth and Mortality Rates in Three Pine Species under Mediterranean Conditions

Cristina Valeriano [1,2], Antonio Gazol [1], Michele Colangelo [1,3] and Jesús Julio Camarero [1,*]

1 Instituto Pirenaico de Ecología (IPE-CSIC), 50192 Zaragoza, Spain; cvaleriano@ipe.csic.es (C.V.); agazolbu@gmail.com (A.G.); michelecolangelo3@gmail.com (M.C.)

2 Departamento de Sistemas y Recursos Naturales, Universidad Politécnica de Madrid, Ciudad Universitaria, 28040 Madrid, Spain

3 School of Agricultural, Forest, Food and Environmental Sciences (SAFE), University of Basilicata, 85100 Potenza, Italy

* Correspondence: jjcamarero@ipe.csic.es; Tel.: +34-976-363-222 (ext. 880041)

**Abstract:** Drought constrains tree growth in regions with seasonal water deficit where growth decline can lead to tree death. This has been observed in regions such as the western Mediterranean Basin, which is a climate-warming hotspot. However, we lack information on intra- and interspecific comparisons of growth rates and responses to water shortage in these hotspots, considering tree species with different drought tolerance. We sampled several sites located in north-eastern Spain showing dieback and high mortality rates of three pine species (*Pinus sylvestris, Pinus pinaster, Pinus halepensis*). We dated death years and reconstructed the basal area increment of coexisting living and recently dead trees using tree ring data. Then, we calculated bootstrapped Pearson correlations between a drought index and growth. Finally, we used linear mixed-effects models to determine differences in growth trends and the response to drought of living and dead trees. Mortality in *P. sylvestris* and *P. pinaster* peaked in response to the 2012 and 2017 droughts, respectively, and in sites located near the species' xeric distribution limits. In *P. halepensis*, tree deaths occurred most years. Dead trees showed lower growth rates than living trees in five out of six sites. There was a strong growth drop after the 1980s when climate shifted towards warmer and drier conditions. Tree growth responded positively to wet climate conditions, particularly in the case of living trees. Accordingly, growth divergence between living and dead trees during dry periods reflected cumulative drought impacts on trees. If aridification continues, tree drought mortality would increase, particularly in xeric distribution limits of tree species.

**Keywords:** drought stress; forest dieback; *Pinus halepensis* Mill.; *Pinus sylvestris* L.; *Pinus pinaster* Ait.; tree mortality

## 1. Introduction

Climate warming is turning drought into a global constraint of forest productivity and tree growth [1]. Increasing aridification is negatively impacting forest health [2]. This is due to the limiting effects of rising air temperatures and increasing vapor pressure deficit on radial growth and carbon uptake by enhancing water loss through stomata and uncoupling root water and nutrient uptake from dry soils [3–5]. Water shortage limiting photosynthesis rates or constraining transpiration and growth may cause hydraulic failure and tree death through fast dehydration of tissues [6]. Such hydraulic failure and the inability to uptake soil water through roots can lead to meristem impairment and tree death [6]. Drought-induced mortality can also lead to shifts in forest composition by accelerating successional dynamics towards drought-tolerant communities and by rising mortality rates of drought-sensitive species [7–10]. Ideal areas for studying drought impacts on forests are regions subjected to variable precipitation amounts and steadily rising temperatures.

Some regions of Southern Europe such as the Iberian Peninsula are witnessing drought impacts on forests through increased competition for soil water [11] and rising mortality rates [12]. In these regions, there is strong evidence that drought severity, i.e., the negative impact of water shortage on forest ecosystems, is increasing due to the temperature rise and the amplification of water demand [13]. Given that summer temperatures have increased at unprecedented rates in this region [14], and that drought events will become more frequent and severe there [15], understanding how conifer forests respond to hotter droughts (*sensu* [2]) becomes a paramount topic in forest research.

Severe droughts during the warmest decades of the late 20th and early 21st centuries have been linked to forest dieback episodes affecting several conifers [16]. Both pine species with Eurasian distribution such as Scots pine (*Pinus sylvestris* L.) and lowland Mediterranean pine species such as Aleppo pine (*Pinus halepensis* Mill.), which show high resilience [17], have been impacted by these droughts showing growth declines and canopy dieback in southern Europe [16–19]. In addition, other Mediterranean pine species such as black pine (*Pinus nigra* Arn.) [20,21] and maritime pine (*Pinus pinaster* Ait.) also presented dieback symptoms (growth reduction, leaf shedding, mortality of twigs and branches in the canopy, etc.) in several south-European regions after dry spells [22–26].

Growth reductions in response to drought have been widely described in conifers [17,26]. However, we lack robust inter-specific comparisons among pine species showing different drought tolerance, presenting high mortality rates as a result of recent dry spells, and inhabiting sites with contrasting climate conditions across water availability gradients. Not all trees respond to drought in a similar way and so coexisting individuals may be resilient or die depending on local factors such as tree size, soil conditions and past management legacies, among others [27,28]. Along with this, it is certainly unknown when a drought might surpass a lethal threshold leading to growth decline and tree death [29,30]. Drought severity or duration may surpass a certain physiological limit in which trees can no longer tolerate leading to growth decline prior to tree death [16]. Thus, additional comparisons of individuals within species are paramount to forecast which pine individuals, populations and species are more vulnerable to drought damage. It may be expected that drought differently affects pine species at drier versus wet sites and that Eurasian species (e.g., *P. sylvestris*) will be more impacted than drought-tolerant Mediterranean species (e.g., *P. halepensis*).

Combining proxies of tree vigor such as canopy transparency with tree-ring allow quantifying the impacts of past drought on growth of coexisting living and dead trees [16,31]. For instance, resilience of trees which are prone to die (hereafter "dead" trees) is lower than that of living trees [32,33]. Further, it is possible to identify if intrinsic tree features such as size [34] or age [35] determine growth responses to droughts. Ultimately, discerning the potential impact of intrinsic tree features may facilitate the development of further studies to disentangle how other extrinsic factors modulate drought-induced forest mortality events.

Here we compare the growth rates of living and recently dead trees in stands of three pine species (*P. sylvestris*, *P. pinaster* and *P. halepensis*) showing drought-triggered high mortality and severe canopy dieback symptoms in north-eastern Spain. We compare these three species because they encompass wide climatic and ecological gradients considering European pines, from mountain mesic sites (*P. sylvestris*) to dry mid-elevation (*P. pinaster*) and semi-arid lowlands (*P. halepensis*) [36]. Our hypothesis is that living trees will show higher growth rates and lower responsiveness to drought than dying and recently dead trees, whose growth should be more constrained by recent severe water shortage.

Here we aim: (i) to determine the time of tree death using annual rings, (ii) to assess how mortality was related to drought, (iii) to reconstruct and compare radial growth rates of coexisting living and dead trees, and (iv) to determine if dead trees were more responsive to drought than living conspecifics. Obtaining this information would advance our understanding on where and when severe dieback develops in response to drought occurs leading to non-linear responses such as high damage and mortality rates, and a loss of productivity and growth at local scale.

## 2. Materials and Methods

### 2.1. Study Sites

We selected eight sites located in north-eastern Spain (Figure S1) showing recent signs of extensive canopy dieback and elevated mortality rates linked to recent droughts (Table 1). The study forests are dominated by three different pine species: *Pinus sylvestris* L., *Pinus pinaster* Ait., and *Pinus halepensis* Mill. We sampled from two to three sites per species to consider variability among sites and tree populations.

**Table 1.** Characteristics of the study sites. Climate data correspond to the 1970–2020 period.

| Species | Site (Code) | Latitude N | Longitude (−W, +E) | Elevation (m a.s.l.) | Temperature (°C) | Precipitation (mm) | Undergrowth Species [1] |
|---|---|---|---|---|---|---|---|
| *Pinus sylvestris* | Hereña (HE) | 42.77 | −2.92 | 580 | 11.0 | 852 | Qf, Qi, Jp |
| | Calomarde (CA) | 40.37 | −1.56 | 1340 | 12.2 | 390 | Qf, Jt, Jo |
| | Corbalán (CO) | 40.28 | −0.78 | 1303 | 12.6 | 421 | Pn, Qf, Qi, Jp |
| *Pinus pinaster* | Orera (OR) | 41.31 | −1.45 | 884 | 13.2 | 405 | Qi |
| | Miedes (MI) | 41.27 | −1.43 | 963 | 12.8 | 418 | Qi, Au |
| | Mora de Rubielos (MR) | 40.23 | −0.73 | 1186 | 12.2 | 462 | Qi, Pn, Jp, Jo |
| *Pinus halepensis* | Sierra de Luna (SL) | 41.98 | −0.84 | 493 | 13.4 | 485 | Jo, Jp, Qc, Pl |
| | Peñaflor (PE) | 41.78 | −0.72 | 375 | 15.5 | 376 | Jp, Jt |

[1] Species' abbreviations: Au, *Arctostaphylos uva-ursi*; Jo, *Juniperus oxycedrus*; Jp, *Juniperus phoenicea*; Jt, *Juniperus thurifera*; Pl, *Pistacia lentiscus*; Pn, *Pinus nigra*; Qc, *Quercus coccifera*; Qf, *Quercus faginea*; Qi, *Quercus ilex*.

Study sites are subjected to Mediterranean and continental climate with mean temperatures ranging 11.0–15.5 °C and annual precipitation 376–852 mm (Table 1). The coolest-wettest sites are dominated by Scots pine, whereas the warmest-dries sites correspond to Aleppo pine (Figure S1). The estimated annual water balance (difference between precipitation and potential evapotranspiration, which was calculated using the Hargreaves–Samani method) may vary from −470 mm in the Aleppo pine sites to −210 mm in the driest Scots pine sites (Calomarde and Corbalán) and 320 mm in the wettest Scots pine site (Hereña). The three species form pure or mixed stands, mainly with oak or beech species such as *Quercus coccifera* L., *Quercus ilex* L., and *Fagus sylvatica* L. in the case of *P. halepensis*, *P. pinaster*, and *P. sylvestris*, respectively. The understory is dominated by oak and juniper species (see Table 1).

The selected tree species are shade-intolerant, evergreen pine species showing isohydric behavior (strong regulation of leaf water potential regardless environmental conditions and closing stomata under drought) and inhabiting sites with different climatic water balance, from mountain mesic sites (*P. sylvestris*, mean annual water balance of −16 mm) to low- and mid-elevation Mediterranean mountain sites (*P. pinaster*, mean annual water balance of −546 mm) and semi-arid sites (*P. halepensis*, mean annual water balance of −627 mm) [17].

Soils are loamy of the cambisol (Corbalán Scots pine site) and regosol (Aleppo pine sites) types. Soils are basic in all sites excepting the three maritime pine sites (Orera, Miedes and Mora de Rubielos) where acid sandy or clay soils develop on slate and quartzite substrates. Some gypsum substrate may appear in the Aleppo pine sites.

Sampled stands have not been managed since the 1950s. Tree age data suggest sampled stands are naturally established, but the presence of logged stems in the Hereña Scots pine site and its low age variability indicate it could be an old, unmanaged plantation. The small size, and low age of Sierra de Luna *P. halepensis* trees suggests this was a post-fire young stand (Table 2).

**Table 2.** Structural variables measured or calculated for the three pine species. Basal area and density refer to both living and dead trees. Dbh is the diameter at breast height.

| Tree Species | Site | Basal Area (m² ha⁻¹) | Density (Ind. ha⁻¹) | Mortality (%) | Dbh (cm) | No. Sampled Dead/Living Trees | Age at 1.3 m (years) |
|---|---|---|---|---|---|---|---|
| *P. sylvestris* | HE | 4.26 | 799 | 25.0 | 20.5 ± 1.7 | 2/30 | 107 ± 2 |
| | CA | 6.20 | 784 | 37.1 | 25.0 ± 1.5 | 9/13 | 133 ± 5 |
| | CO | 4.00 | 126 | 95.0 | 27.3 ± 1.0 | 38/57 | 138 ± 6 |
| *P. pinaster* | OR | 6.32 | 779 | 50.0 | 28.3 ± 0.9 | 17/16 | 87 ± 2 |
| | MI | 5.73 | 791 | 41.2 | 21.7 ± 1.3 | 15/15 | 82 ± 2 |
| | MR | 2.05 | 608 | 21.1 | 13.9 ± 1.7 | 2/22 | 85 ± 6 |
| *P. halepensis* | SL | 2.12 | 577 | 26.3 | 12.8 ± 2.4 | 13/13 | 53 ± 3 |
| | PE | 4.50 | 179 | 44.0 | 32.3 ± 1.4 | 26/14 | 78 ± 5 |

*2.2. Field Sampling*

At each site, a 50 × 10 m rectangular plot was located in a representative place with high dieback and mortality rates. The mortality of trees due to drought was recognized as a rapid shoot death and leaf loss leading to complete defoliation and canopy dieback of isolated trees and small tree patches. We assumed such mortality hotspots correspond to the impact of recent droughts since pathogens or pests were not observed and most dead trees were located on xeric sites with southern exposure, at low elevation and on rocky, shallow soils. We only found some mistletoe (*Viscum album* L.) presence in the Peñaflor (PE) Aleppo pine site.

We noted or measured the following variables in all trees found within each plot: tree species, Diameter at breast height (Dbh), and canopy defoliation (%). Dead trees were those showing complete defoliation (100%) or just retaining dead branches and needles but preserving bark and thick branches. Living trees always presented defoliation lower than 50%. We took two cores at 1.3 m perpendicular to the maximum slope in mature living and dead trees of the dominant pine species using Pressler increment borers (Haglöf Sweden, Sweden). Dead and living trees are abbreviated as D and ND trees hereafter.

*2.3. Dendrochronological Analyses*

In total, a balanced number of mature D and ND trees were cored for dendrochronological purposes in most sites over the years 2019 and 2020 (Table 2). Cores were air dried and carefully sanded to distinguish the rings following standard procedures in dendrochronology [37]. Tree samples were visually cross dated using marker rings [38]. For dead trees the outermost ring on cross-dated samples was considered as the year in which a tree died [39]. Then, the tree-ring widths (TRW) were measured with a 0.001 mm resolution on images obtained in a scanner (Epson Expression 10000XL) and using the CDendro and CooRecorder software [40]. The visual cross-dating was checked using the software COFECHA software (ver. 6.06P, Laboratory of Tree-Ring Research, The Univ. of Arizona, AZ, USA), which calculates shifting correlations with a mean site series [41]. We also calculated several statistics to characterize these series such as the mean tree-ring width, the mean sensitivity or relative difference in width between consecutive rings, the mean of correlations among all radii (rbar) and the expressed population signal (EPS) which measures how coherent and replicated is a chronology [37]. A well-replicated period was considered when EPS > 0.85 (cf. [42]). Finally, TRW measurements were transformed into basal area increments (BAI) by assuming a circular shape of stems as:

$$\text{BAI} = \pi \left( R^2_t - R^2_{t-1} \right) \tag{1}$$

where *R* is the tree radius and *t* the year of the ring growth. We obtained standardized and detrended BAI series by fitting spline functions to raw BAI series in order to remove the age effect. The detrending of BAI data was performed using the R package dplR [43].

### 2.4. Drought: Spatial and Temporal Patterns

To quantify and characterize drought, we used the Standardized Precipitation Evapotranspiration Index (SPEI) [44]. The data series of SPEI were obtained from a Spanish SPEI database gridded at 1.1-km$^2$ with a weekly temporal resolution for each site and considering the period 1970–2020 [45]. The SPEI was calculated at three-time resolutions (3, 6, 9 and 12 months) to inspect their impacts on growth. We also obtained monthly values of mean temperature and total precipitation corresponding to the same 1.1-km$^2$ database.

To determine how extreme recent droughts were, first we obtained the whole dataset of 12-month August SPEI data for the period 1970–2020. Based on a prior study [46], we selected the 12-month August SPEI since it accounted for a high amount of growth variability. Second, we focused on the most extreme droughts since 2010 which preceded dieback events in Spain (cf. [16,47]), namely 2012 and 2016–2017 (hereafter 2017 drought). The SPEI values were obtained for the 1.1-km$^2$ grids where each pine species was present according to the 2nd and 3rd Spanish National Forest Inventories (carried out in 1986–2007) and for the eight study sites considering the period 1970–2020. The Spanish National Forest Inventory consists of circular plots of 25-m radius systematically distributed on a 1-km$^2$ cell grid over forested areas [48]. Then, we compared the species' distribution SPEI values with those of the study sites to evaluate how extreme were the 2012 and 2017 droughts. Other severe droughts occurred in 1994–1995 and 2005, but we did not consider them since it was not possible to sample dead trees whose death occurred before 2010.

### 2.5. Drought-Growth Relationships

To determine the association among growth rates (detrended BAI series) and drought (SPEI) we calculated bootstrapped Pearson correlations considering the period 1970–2020. Detrended BAI series and monthly SPEI values obtained at 3-, 6-, 9- and 12-month scales from the previous October to current December were analyzed by using the R package *Treeclim* [49].

### 2.6. Statistical Analyses

Since there were few D trees sampled in Hereña and Mora de Rubielos (Table 2), we only considered the six remaining sites in the following models of growth. We used linear mixed effects model [50] to test those particular time periods in which coexisting living and dead trees showed differences in growth [51]. Models were fitted in the R statistical framework [52] using the *nlme* package [53]. The analyses were performed by following three consecutive steps. First, we fitted a model in which growth (log-transformed $BAI_{ij}$ + 1) was modeled as a function of tree status (ND, D) and time (years):

$$\text{Log } (BAI_{ij} + 1) = b0 + u0_j + X_{ij} \, b + \varepsilon_{ij} \tag{2}$$

In this model, intercepts of the average fixed effect and random effects are represented by b0 and $u0_j$. The matrix of explanatory variables is $X_{ij}$ and $\varepsilon_{ij}$ is the error for subject j at time i. Tree identity (j) was regarded as random factor since ring-width series represent repeated measures over the same individual.

Second, we fitted autoregressive moving average models (ARIMA) with structure (1, 0, 1) to residuals of the linear mixed-effect model ($\varepsilon_{ij}$) to eliminate temporal autocorrelation. Third, we used Student's t-tests to check if means of the fitted residuals differed from zero for each year to find differences annual growth between ND and D trees for the period 1950 to 2019. The analyses were carried out using the *glmmTMB* and forecast R packages [54,55].

We used again linear mixed effects models to determine differences in growth trends and the response to drought of D and ND trees for the common period 1970–2012. In this case, log-transformed BAI was modelled as a function of tree Dbh, tree status (ND vs. D), year and the 12-month SPEI for August. Two interactions (status × year, status × SPEI) were considered to test for different response between ND and D trees:

$$\text{Log (BAI}_{ij} + 1)\sim 1 + \text{year}_j + \text{Dbh}_j + \text{status}_j + \text{SPEI}_{ij} + \text{status}_j \times \text{year}_j + \text{status}_j \times \text{SPEI}_{ij} + \text{AR(1)} + 1 \mid \text{site:j} \qquad (3)$$

Separate analyses were performed for each site considering tree identity as random factor (j) and accounting for first-order autocorrelation (AR(1)) of growth. A global model was fitted considering site and tree identity as random factors.

Maximum likelihood was used to estimate the coefficients of the models and we selected the model showing the minimum value of the corrected Akaike information criterion (AICc) [56]. Differences in fixed factors between D and ND were tested using post-hoc analyses. We estimated the marginal means for the two interactions of the selected model by using the *emmeans* packages [57]. Model selection was done using the *MuMIn* R package [58]. Finally, the explanatory power of selected models was quantified by calculating the conditional ($R^2c$) and marginal ($R^2m$) coefficients of determination which account for the effects of fixed and fixed plus random effects, respectively [59].

## 3. Results

### 3.1. Drought Severity and Mortality Patterns

The 2005 and 2012 droughts were among the most extremes for the three study species, whereas the 2017 drought was extreme for *P. pinaster* (Figure 1). Accordingly, mortality peaked in 2012 in the case of the Corbalán *P. sylvestris* and Peñaflor *P. halepensis* sites. In 2017, there was a peak in the number of deaths for both *P. pinaster* sites, followed by high mortality rates in 2018 (Corbalán *P. sylvestris* site), 2019 (Orera and Miedes *P. pinaster* sites) and 2020 (Calomarde *P. sylvestris* site, Sierra de Luna *P. halepensis* site) (Figure 2).

Mortality rates ranged between 21% (Mora de Rubielos *P. pinaster* site) and 95% (Corbalán *P. sylvestris* site; Table 2). The site mortality rate was positively related to the basal area (Pearson r = 0.964, $p < 0.001$). The recruitment distributions did not significantly differ (Kolmogorov-Smirnov tests, $p > 0.05$) between ND and D trees indicating they corresponded to similar cohorts (Figure S2). The sampled trees corresponded to mature stands (age ranges between 73 and 144 years), excepting the Sierra de Luna *P. halepensis* young stand where trees were around 50–60 years old (Table 2).

We found higher growth rates (BAI) in living (ND) than in dead (D) trees in all sites excepting the Orera *P. pinaster* site (Figure 3 and Table 3). In all sites, BAI decreased during dry periods or years such as the 1990s, 2005, 2012 and 2017, and increased during wet decades such as the 1970s (Figure 3 and Figure S3). Significant differences in BAI between ND and D trees were mainly found for years after the 1970s, excepting in the Calomarde *P. sylvestris* site (Figure 3, Table S1). The longest period with a BAI difference between ND and D trees were found in the Orera *P. pinaster* and the Corbalán *P. sylvestris* sites, albeit with opposite signs. In Orera, recently dead trees grew more in the past, but in Corbalán dead tree showed persistently lower growth rates indicating a chronic dieback phenomenon. In the two *P. halepensis* sites, the growth divergence between D and ND trees was mainly observed during dry periods such as the 1980s, 1990s and 2000s. A similar pattern, was restricted to shorter periods or specific droughts (e.g., 2005, 2012, 2015), was also observed in the Calomarde *P. sylvestris* and Miedes *P. pinaster* sites.

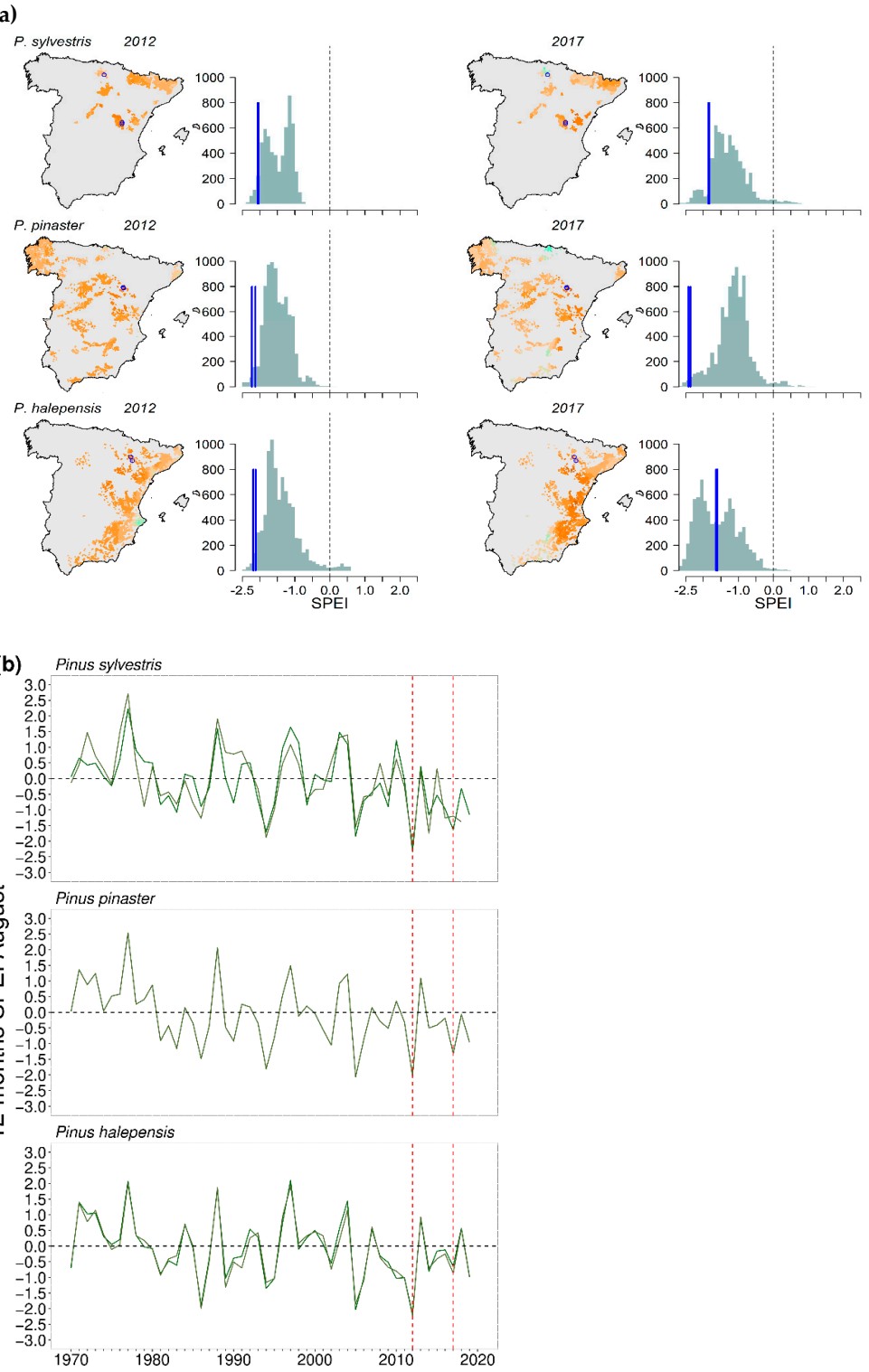

**Figure 1.** Spatial (**a**) and temporal (**b**) patterns of drought severity based on 12-month SPEI August values. The histograms (**a**) show the distribution of SPEI values for the 2012 and 2017 droughts as compared with SPEI values of the whole distribution area of the three study pine species in Spain (maps with orange patches and symbols showing sites' locations). The lower plot (**b**) shows the mean SPEI values for each species considering the period 1970–2019 (in the case of *P. sylvestris* and *P. halepensis* two curves are presented corresponding to two 1.1-km² grids). The vertical dashed lines indicate the 2012 and 2017 droughts.

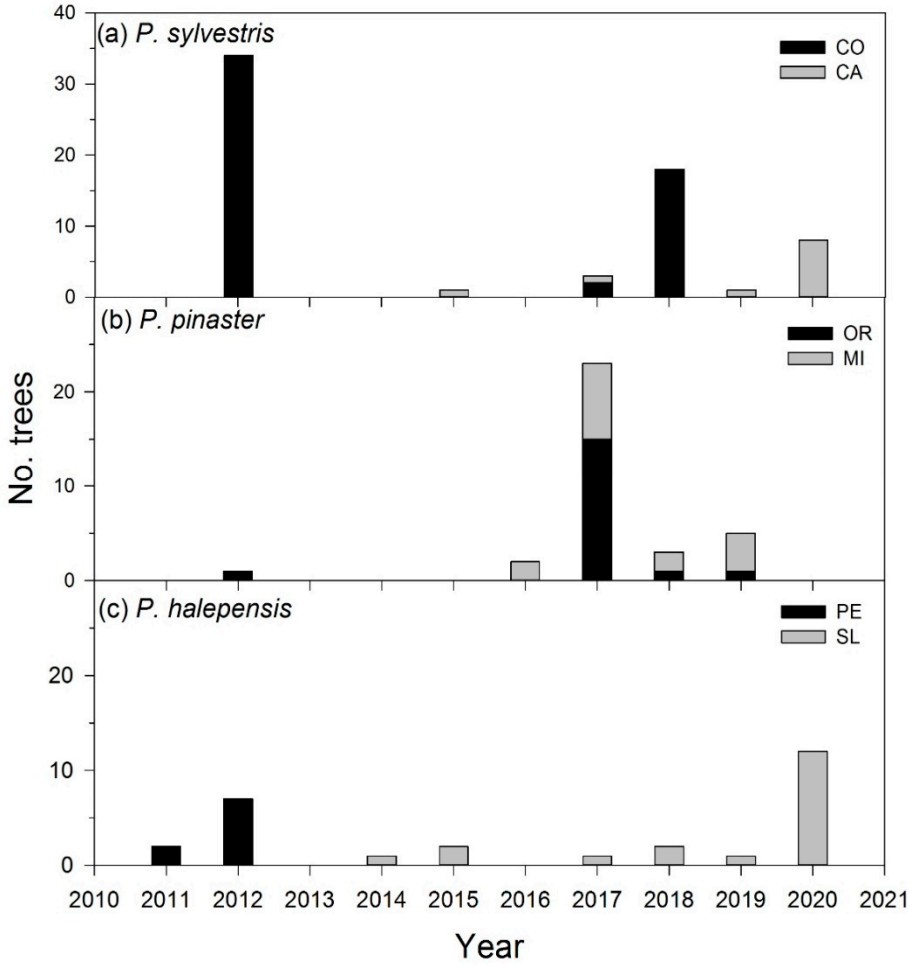

**Figure 2.** Frequency of recent tree deaths observed in the three study species (**a–c**) and considering two sites per species (see Table 1). Sites' codes are in Table 1.

**Table 3.** Growth data and related statistics. Values are means ± SD. The rbar is the mean of correlations among all radii, and the EPS is the Expressed Population Signal. Different letters indicate significant (*p* < 0.05) differences in mean basal area increment or tree-ring width between living (ND) and dead (D) trees according to Student's *t*-tests.

| Species | Site | Status | Basal Area Increment (cm²) | Tree-Ring Width (mm) | First-Order Autocorrelation | Mean Sensitivity | rbar | Period with EPS > 0.85 |
|---|---|---|---|---|---|---|---|---|
| *P. sylvestris* | CA | ND | 2.91 ± 0.18 | 0.88 ± 0.50 | 0.63 | 0.371 | 0.51 | 1845–2019 |
| | | D | 3.54 ± 0.20 | 0.76 ± 0.46 | 0.66 | 0.400 | 0.50 | 1895–2019 |
| | CO | ND | 3.11 ± 0.17 | 1.01 ± 0.62b | 1.01 | 0.436 | 0.56 | 1844–2019 |
| | | D | 3.80 ± 0.14 | 0.71 ± 0.47a | 0.60 | 0.492 | 0.62 | 1838–2012 |
| *P. pinaster* | OR | ND | 4.14 ± 0.21 | 1.24 ± 1.11 | 0.73 | 0.472 | 0.60 | 1920–2019 |
| | | D | 4.85 ± 0.21 | 1.25 ± 1.00 | 0.72 | 0.452 | 0.63 | 1917–2016 |
| | MI | ND | 2.10 ± 0.11 | 0.88 ± 0.63 | 0.61 | 0.429 | 0.60 | 1921–2020 |
| | | D | 1.52 ± 0.07 | 0.74 ± 0.61 | 0.71 | 0.461 | 0.54 | 1921–2020 |
| *P. halepensis* | PE | ND | 2.74 ± 0.13 | 1.08 ± 0.66 | 0.52 | 0.442 | 0.50 | 1946–2020 |
| | | D | 2.17 ± 0.12 | 0.97 ± 0.68 | 0.65 | 0.447 | 0.49 | 1916–2015 |
| | SL | ND | 2.34 ± 0.12 | 1.13 ± 0.78 | 0.60 | 0.480 | 0.48 | 1969–2019 |
| | | D | 1.64 ± 0.12 | 0.96 ± 0.77 | 0.76 | 0.486 | 0.39 | 1966–2015 |

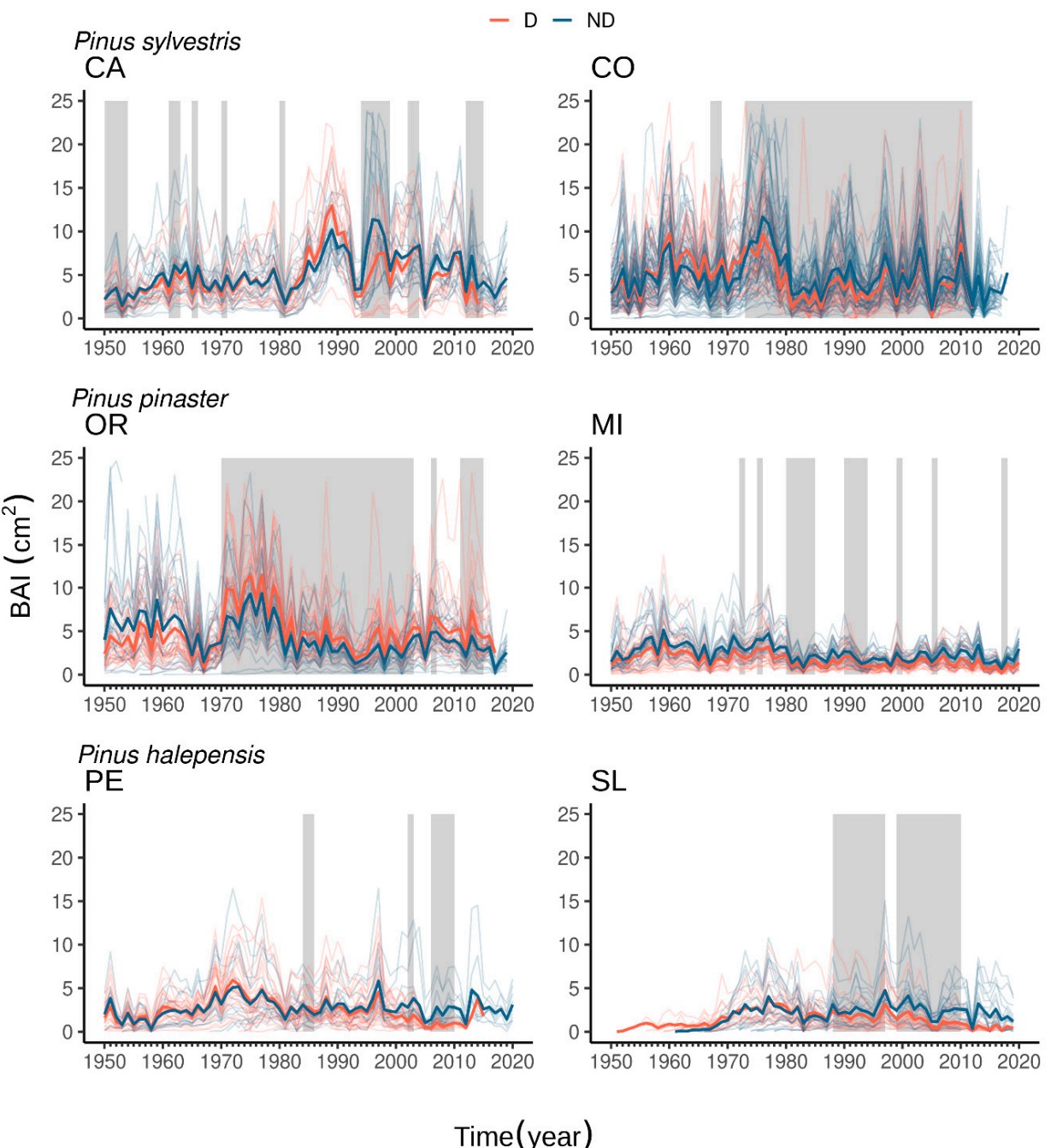

**Figure 3.** Basal area increment (BAI) series obtained for dead (D, red lines) and living (ND, blue lines) trees in six study sites (Table 3). Solid and thicker lines show the mean BAI data of each group. The grey boxes show periods with significant (*p* < 0.05) differences in BAI between D and ND trees (see Table S2).

### 3.2. Drought-Growth Relationships

The 12-month SPEI values from July to September showed high positive Pearson correlations with detrended BAI series for all sites and species (Figure S4). The highest correlations were observed for the *P. pinaster* sites and the Peñaflor *P. halepensis* site, whereas the lowest correlations were found for the Calomarde *P. sylvestris* site.

The linear mixed effects models showed significant negative effects of tree status (ND vs. D trees), a positive effect of 12-month SPEI August, and mixed effects of the status x SPEI interaction on BAI in all sites, excepting the Calomarde *P. sylvestris* and the Peñaflor *P. halepensis* sites (Table 4, Tables S3 and S4). The proportion of BAI variance accounted

for by fixed factors ($R^2$m) was relatively high in the two *P. pinaster* sites (34.3–36.5%) and low in the Sierra de Luna *P. halepensis* site (14.9%). This value was also low for the global model (11.6%), which showed significant effects on BAI of tree Dbh and status, SPEI and the interactions SPEI × status and Year × status.

**Table 4.** Results of the linear mixed-effects models fitted to basal area increment data. The values show the estimated effects of each factor and its interactions. Tree status corresponds to living and dead trees. The last line corresponds to the global model fitted to all sites. See sites' codes in Table 1.

| Species | Site | Intercept | Year | Dbh | Status | SPEI | SPEI × Status | Year × Status | Akaike Weight | $R^2$m (%) | $R^2$c (%) |
|---|---|---|---|---|---|---|---|---|---|---|---|
| *P. sylvestris* | CA | −1.252 | 0.001 | 0.011 | −6.183 | 0.091 | – | 0.003 | 0.67 | 17.7 | 37.1 |
| | CO | 5.147 | −0.002 | – | −2.659 | 0.127 | −0.012 | 0.001 | 0.97 | 20.8 | 59.6 |
| *P. pinaster* | OR | 5.924 | −0.002 | 0.015 | −0.089 | 0.103 | −0.015 | – | 0.66 | 34.3 | 62.2 |
| | MI | 5.500 | −0.003 | 0.016 | −1.746 | 0.058 | 0.014 | 0.001 | 0.99 | 36.5 | 59.7 |
| *P. halepensis* | PE | 20.180 | −0.010 | – | – | 0.063 | – | – | 0.21 | 29.7 | 66.4 |
| | SL | 10.736 | −0.005 | – | −14.143 | 0.049 | 0.024 | 0.007 | 0.53 | 14.9 | 44.4 |
| All sites | – | – | 7.025 | −0.003 | 0.011 | −5.967 | 0.067 | 0.012 | 0.003 | 0.99 | 11.6 | 61.4 |

We found differences between ND and D trees in the year and SPEI effects on BAI in several sites, excepting in the Peñaflor *P. halepensis*, the Miedes *P. pinaster* and the Calomarde *P. sylvestris* sites (Figure 4). The strongest positive effect of SPEI on BAI was observed in the Corbalán *P. sylvestris* site. The BAI × SPEI interactions were strong in Corbalán *P. sylvestris*, Miedes *P. pinaster* and Sierra de Luna *P. halepensis* sites (Table S3).

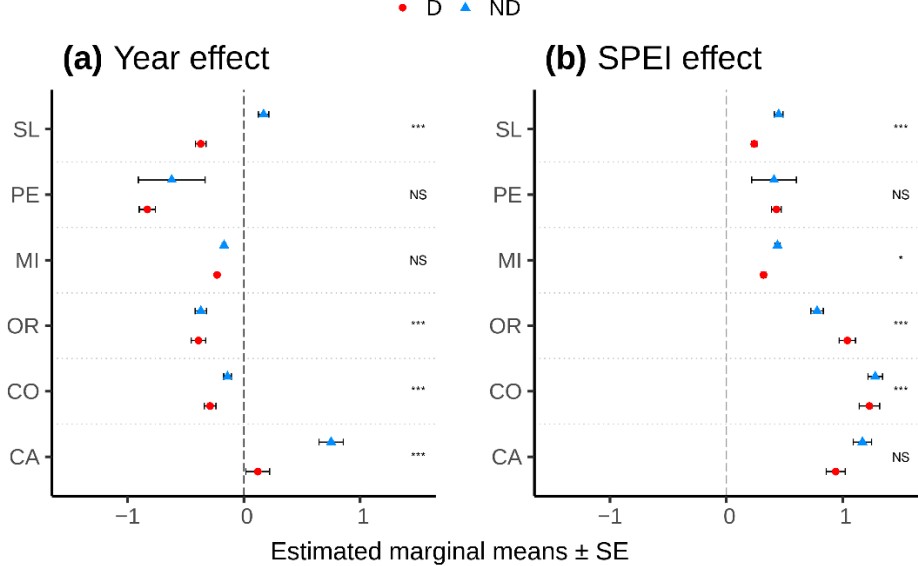

**Figure 4.** Estimated effects of (**a**) Year and (**b**) SPEI on growth depending on tree status (D, dead trees; ND, living trees). Values are estimated marginal means with standard errors (SE) according to the linear mixed-effect models. Significance levels: NS, not significant; * *p* <0.05, *** *p* < 0.001. See sites' codes in Table 1.

The models fitted to BAI showed that the rate of growth reduction as SPEI increases was higher in D than in ND, whilst the trend in BAI of D trees decreased more steeply after the 1970s than in the case of ND trees (Figure 5).

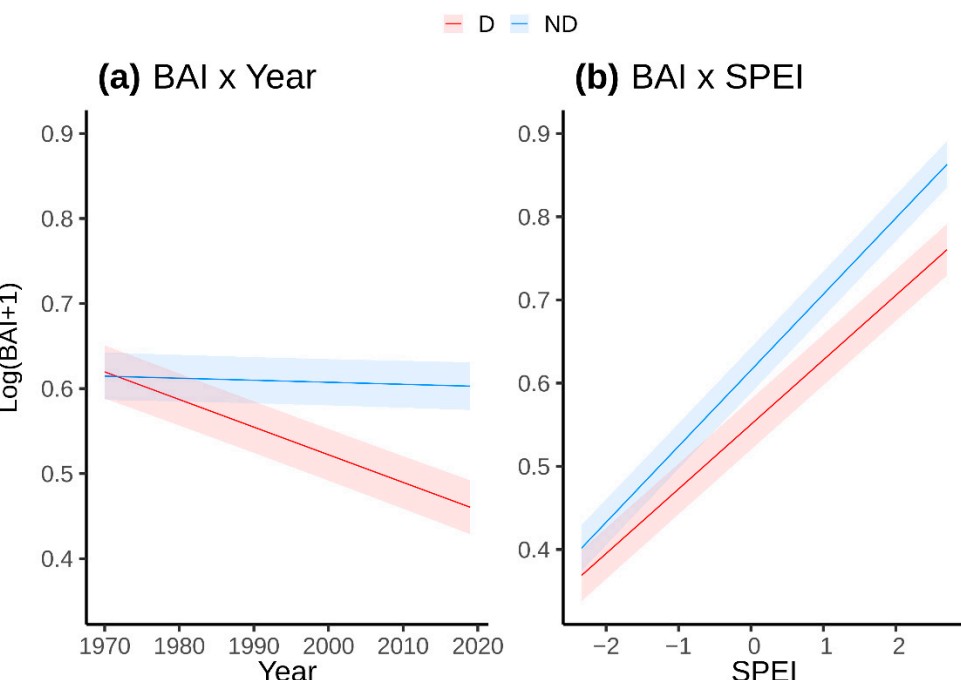

**Figure 5.** Modelled interactions on basal area increment as a function of (**a**) time and (**b**) the SPEI drought index for dead (D, red lines and areas) and living (ND, blue lines and areas) trees. Lines show the estimated effects, and the areas show standard errors.

## 4. Discussion

Recent droughts (2012, 2017) triggered mortality pulses (Figure 2) in the three study pine species which were linked to previous growth declines and recent, massive canopy dieback (Figure S1). As expected, living trees showed higher growth rates than dead trees, excepting in the Orera *P. pinaster* site (Figure 3). Unexpectedly, recently dead trees tended to show lower growth responsiveness to drought severity when climate conditions are wet (SPEI > 0) but presented a similar responsiveness as living trees for dry (SPEI < −1) climate conditions (Figure 5 and Figure S3).

Mortality rates peaked in 2012 and 2017 in response to two of the most severe droughts affecting north-eastern Spain since 2010 (Figures 1 and 2) [60]. These mortality peaks were more evident in the *P. sylvestris* and *P. pinaster* sites, whereas in the *P. halepensis* sites mortality was steady, particularly in the Sierra de Luna site. Post-drought mortality was also observed in 2018–2020 and such lagged responses have been documented before for *P. sylvestris* [26]. Interestingly, the drought of 2012 was severe for the three species according to the SPEI (Figure 1), but the impact on *P. halepensis* mortality was minor (Figure 2). Similar results were found by Gazol et al. [61] who found a strong drought-induced mortality of *Juniperus phoenicea* in the Monegros steppe (north-eastern Spain), whereas no impacts were observed in *P. halepensis*. However, the 2017 was particularly severe, in terms of low SPEI values and high mortality rates, for *P. pinaster*. The 2012 drought started in the prior 2011–2012 winter and intensified during the 2012 spring, severely impacting *P. sylvestris*, which depends on soil water replenishment in winter [16]. However, the 2017 drought started in late early summer which could explain why impacted *P. pinaster* sites, since this species depends on adequate soil water supply during summer and autumn [61]. This indicates that seasonality and duration of the drought are also critical features of its impact on tree mortality and growth [46,62]. Site conditions or local climate features (e.g., convective summer storms) could also explain the lower impact of the 2012 drought on *P. pinaster*. These results confirm that tree populations in their xeric limits of distribution are usually very sensitive to drought, but there may be exceptions due to local features [63]. In addition, our findings support forecasts of higher vulnerability of some *P. sylvestris*

populations in the Mediterranean basin in response to a warmer and drier climate leading to increase vapor pressure deficit [64,65].

The results partially support previous analyses of forest inventory data indicating that competition for soil water contributed to explain mortality in Iberian pine forests [11,12]. We found a positive association between recent mortality incidence and basal area. However, our analyses were restricted to eight sites with elevated mortality and intense dieback, so further research could consider dating and analyzing additional sites and species. It is also possible that forests with the worst site quality are characterized by low basal area values and low mortality rates or that basal area has increased in sites with high mortality because of the replacement of drought-sensitive by drought-tolerant tree species [9,66]. In fact, the Corbalán site, which displayed the highest mortality rates, presented a complete lack of *P. sylvestris* recruitment and a shift in species composition towards a higher dominance of more drought-tolerant species such as *P. nigra* or *Q. ilex* [66].

We found strong and long-lasting growth declines for dead trees in most sites, which agree with what Cailleret et al. [32] reported for gymnosperms. The exception to this pattern was the Orera *P. pinaster* site, where the growth of dying trees was more negatively impacted by drought than in living trees (Figure 4). This dieback event could correspond to a higher vulnerability of fast-growing trees linked to a more profuse use of water through the formation of wider conduits which are also more prone to experience xylem embolism [67,68]. A similar pattern was also observed in *Quercur robur* [69] and in *P. sylvestris* [67]. Therefore, these dieback episodes may be predisposed by physiological differences among trees prior to the drought onset. In this respect, *P. sylvestris* is very sensitive to drought and its leaves close stomata rapidly to prevent the development of xylem embolism [70]. Regarding *P. pinaster*, this species increased its water use efficiency in response to water shortage showing a high acclimation capacity [71]. Lastly, *P. halepensis* was able to recover from xylem cavitation and loss of hydraulic conductivity after recurrent droughts showing its high tolerance to seasonal water deficit [72]. In these cases, dying trees underperformed than living trees during dry conditions, in our case from the 1980s onwards, and possibly succumbed because of hydraulic failure [73]. This site represents an ideal setting to investigate if there are anatomical or morphological differences between D and ND trees, thus linking functional traits and tree vulnerability to drought [74].

The growth patterns in the other sites followed expectations, with steeply declining growth rates of dead trees since the 1980s (Figures 3–5), when a climate shift towards drier conditions occurred [75]. A similar pattern of living trees showing the largest growth rates has also been observed in *Pinus ponderosa* trees from California [76]. Higher growth responsiveness to SPEI in living than in dead trees was observed in Corbalán, Miedes and Sierra de Luna sites (Figure 4). It is remarkable the strong growth divergence between the two vigour classes observed in the Sierra de Luna *P. halepensis* site (Figure 4), where most trees were young (Figure S2) and recruited after wildfire. Here, the growth decline of dead trees started in the 1980s as climate started warming and drying (Figure 1), which suggests a lower performance of dying trees because of inherent features making them prone to dieback such as shallow root systems or cumulative drought effects on hydraulic architecture [70,77]. Such dieback event could also be attributed to different microsite conditions. For instance, a higher rock cover and soil stoniness have been shown to mitigate drought stress and increase survival of *P. halepensis* in Israel [27].

Consecutive or recurrent droughts can lead to drought legacies [78] or cumulative stress making trees prone to die [28]. The growth of the three studied species responded to drought but the strength and duration of their legacies differed [17]. This opens the question if early-warning signals can be recovered from tree growth series at the individual scale [16] and upscaled at stand or forest scales using remote-sensing data. For example, drops in wetness indices derived from satellite images occurred during the dry 1990s in the two study *P. pinaster* sites, which suggests these indices could be used as early-warning signals of dieback and tree drought mortality [25]. We argue that further advances should focus on the individual scale to search for similar early-warning signals in tree-ring (width,

density, isotope composition) or wood anatomical variables. This would improve our forecasting ability of which populations or trees will be more prone to succumb or die due to drought. An adaptive silviculture in the Mediterranean region could assist in shaping forests which are less vulnerable to extreme climatic events such as droughts. In these cases, managers should regulate competition and density-dependent effects and enhance structural and functional diversity and soil water uptake by roots [79].

## 5. Conclusions

We dated the years of death and reconstructed growth variability in three pine species showing extensive dieback and high mortality rates after recent and severe droughts. Mortality peaked during and after severe droughts. Some of the most affected sites, such as the Corbalán *P. sylvestris* and the Miedes *P. pinaster* sites, showed persistent, low growth rates, regardless tree vigor. This chronic condition of low growth was observed since the 1980s when climate warmed and become more arid, which suggests these tree populations are vulnerable and may show local extinction processes due to non-linear responses to severe droughts. Recently, dead trees showed lower growth rates than living trees in most, but not all, sites. Growth of living trees responded to wet climate conditions more than in the case of dead trees. Our findings confirm that lower growth and higher mortality rates occur in response to drier and warmer conditions and suggest that some tree populations will experience more extensive dieback and tree drought mortality as climate keeps warming. Studies such as the one presented here are fundamental to identify vulnerable forests and stands. Similar approaches could be sued to identify "dieback hotpots" where non-linear responses to drought involve high damage and mortality rates and a rapid loss of productivity and growth. These stands could be used as sentinels for monitoring the responses of vulnerable forest ecosystems to ongoing climate warming and aridification.

**Supplementary Materials:** The following are available online at https://www.mdpi.com/article/10.3390/f12121700/s1. Table S1: Statistics of fitted linear mixed effects models and periods with significant ($p < 0.05$) differences in basal area increment data between living (ND) and dead trees according to Student's t-tests. Table S2: Statistics of models of basal area increment fitted to each site. The two last columns show the increment in the corrected Akaike information criterion (AICc) and the relative Akaike weight. The symbol "+" indicates that the effect was included in the selected model. Table S3. Estimated differences in the interactions between basal area increment (BAI) and time (year) or the SPEI drought index between living and dead trees in the study sites. The columns show the estimated coefficient, its stand-ard error (SE) and the significance level ($p$). Figure S1: Location of the eight sampled sites (tree icons) in north-eastern Spain (see sites' coordinates in Table 1). Pictures showed (a) dead and (b) surviving *P. pinaster* trees in Orera and Miedes sites and (c) a zoom-in on the study area in Spain. Pine species are defined by labels' colours: *P. sylvestris* blue, *P.pinaster* green, and *P.halepensis* red. Sites' abbreviations are: HE, Hereña; CA, Calomarde; CO, Corbalán; MI, Miedes; OR, Orera; MR, Mora de Rubielos; PE, Peñaflor; and SL, Sierra de Luna. Figure S2: Recruitment histograms of dead and living trees considering 10-year classes. The y-axis represents the number of individuals. Sites are: CA (Calomarde), CO (Corbalán), OR (Orera), MI (Miedes), PE (Peñaflor) and SL (Sierra Luna). Figure S3: Basal area increment (lines) and SPEI values (columns) in the six study sites according to study species. Note that the SPEI values are the same for the two nearby *P. pinaster* sites. Arrows show the 2005, 2012 and 2017 droughts. Sites are: CA (Calomarde), CO (Corbalán), MI (Miedes), OR (Orera), PE (Peñaflor) and SL (Sierra Luna). Figure S4: Pearson correlations (color scale) obtained by relating the detrended basal area increment series and the SPEI drought index in six study sites (x axes). Correlations were calculated from previous October (oct) to current December (DEC) considering 3- (SPEI3), 6- (SPEI6), 9- (SPEI9) and 12-month (SPEI12) SPEI values. Significant correlations ($p < 0.05$) are indicated with a dot (·). Sites are: CA (Calomarde), CO (Corbalán), MI (Miedes), OR (Orera), PE (Peñaflor) and SL (Sierra Luna).

**Author Contributions:** C.V.: Conceptualization; Data curation; Formal analysis; Writing—original draft; Writing—review and editing. A.G.: Conceptualization; Data curation; Formal analysis; Writing—review and editing. M.C.: Data curation; Writing—review and editing; J.J.C.: Conceptualization; Data curation; Funding acquisition; Writing—review and editing. All authors have read and agreed to the published version of the manuscript.

**Funding:** This research was funded by project FORMAL (RTI2018-096884-B-C31). C.V. acknowledges funding by an FPI fellowship (PRE2019-089800), Spanish Ministry of Science and Innovation.

**Institutional Review Board Statement:** Not applicable.

**Informed Consent Statement:** Not applicable.

**Data Availability Statement:** The original contributions presented in the study are included in the article/Supplementary Material, further data will be made available upon reasonable request to the corresponding author.

**Conflicts of Interest:** The authors declare no conflict of interest. The funders had no role in the design of the study; in the collection, analyses, or interpretation of the data; in the writing of the manuscript; in the decision to publish the results.

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
