# Peer review of "Drought Drives Growth and Mortality Rates in Three Pine Species under Mediterranean Conditions"

_forests, doi:10.3390/f12121700_

Round 1

Reviewer 1 Report

General comments

I have read the manuscript (forests -1481157). Entitle: Drought drives growth and mortality rates in three pine species under Mediterranean conditions written by Cristina Valeriano et. al., for publication of forests MDPI. In this study, the author investigates the Mortality and growth rate of three pine species in could of years’ research (2012-2017). Author found that the death tree found that slow growth. Author also found that the when climate is shifted to drier and warmer the growth response of the tree more positively to wet climates conditions in living than the dead trees. Author also found that the mortality is increased if acidification continues.

The overall research is well conducted and research is obvious application potential for the readers because this study reflected the cumulative drought impact on tree functional mortality. In this sense, the manuscript is much valuable. However, I found some points especially the flow of the text is not smooth and sometimes I found the shallow writing and lack of potential references, and lack of connection of story in different paragraphs especially in the introduction and discussion sections. In discussion, the author should be deal with the physiological perspectives and what will affect plant biology. I also found the lack of potential and appropriate references to support the findings. The author should provide enough examples and their interpretation of different traits of physiological and biochemical responses. I mention some tips and recommended some literature this will help to improve this manuscript quality better than before. Overall after I evaluate this manuscript, I request the author for the “MAJOR REVISION” and also, I request to authors for revision according to the rules of the journal and correct the bibliography.

 Major suggestions

1) Abstract Issue: The author wrote the important finding in their abstract but the text seems like comparatively more description. Please concise the text little bit more with only focus the major outcomes. Abstract should more logical, short, concise, and informative and it should reflect your study and major findings while shortly observed by readers. Please make the necessary corrections. Also please follow the pattern and author pack and previously issued articles, I think the separate “section title” is not needed just author may write thoroughly without separation of subtitle in the abstracts.

2) Introduction: I satisfy with the introduction except the starting parts. Please improve your first and second paragraph more logically. Author should to be integrate the drought component and negative impact in plant biology, however author directly jump the part of rising temperature, water scarcity and then tree mortality. For the sequential flow and for the enough background author should deal the negative consequences of drought stress in this climate change era. The negative effect of drought for the plant biological points of should present with the enough reference before directly go to the tree death part. Please read and cite article entitle: “Response of drought stress in prunus sargentii and larix kaempferii ...https://doi.org/10.1016/j.foreco.2020.118099” and mention that “drought reduced the morphological and physiological traits, reduce the leaf water potential and sap movement due to alternation of xylem anatomical features in the plants”. Then only author should go to the area description and water scarcity scenario (ie. Southern Europe) and thoroughly make the background and raise questions and hypothesis of the study about the forest mortality due to cause of drought.

3) Hypothesis of the study: Author well describe the aim of the study from the line no 88-93, which are more satisfactory. However, author should include the section of the hypothesis of the study as well in the last section of the introduction. Based on the hypothesis author should mentioned the aim of the study. In past paragraph of starting author raise the problem of specific area of Spain and tree mortality and dieback problem due to drought stress which is good text but author also should have mentioned hypothesis of the study for e.g. We hypothesized that these three species are contrasting pattern of physiological and morphological response after exposed to the drought stress therefore…. something like this way. Hypothesis of the study is important thing and it give another strength for the introduction. The hypothesis should be very clear in the introduction sections because, without appropriate literature, questions, or hypotheses in the introduction section the entire text will be unclear. The author should give special attention and the sequential presentation this section in the last part of introduction.

Some others suggestions

4) Table 2 (Line no. 131): Author should full mention their abbreviation if those are used inside the in the footnote, see DBH it probably may be “diameter breast height”. Each table are independent with full information.

5) Materials and methods: Author should be clearly mentioning how author recognize the tree drought severity and its mortality author mentioned somewhat unclear. Please revised this clearly method in MM section because this is main part for your study in different site to identify the drought stress.

6) Figure1, 2, 3…: Please little more increased the font of the letter inside the figures, it seems like difficult to recognized. Also please increased in x-axis and Y axis as well and make this consistence of all the figures.

7) Line no. 322: It is somewhat unclear and also have opposite meaning “living tree showed higher growth rate than dead trees” what you this means. It is always true but I am expecting advance text rather than this. Please revised this.

8) Line no. 328: Author mention that the study region is affected due to drought in the study region (North east Spain) since 2010 and give the citation. That means this type of tree mortality region the study was already conducted before you, if you should have showed the addition thing and should your report differ than that literature.  Please consider this point more seriously.  

 9) Line no. 360-372: Whole of this paragraph in the discussion should be clearer, especially author should focus the line from 363-370. I agree with the author statement that the tree mortality is directly related due to formation of wider conducts that cause make embolism and loss of hydraulic conductivity. In here author should describe more this statement especially anatomical (xylem vessel structure) point of view. Please read this article: Impact of drought stress on photosynthesis responses, leaf water and sap flow…... “DOI:10.1016/j.scienta.2018.11.021 This article describe very well about the hydraulic conductance of the plant and reduction of hydraulic failure and its possible reason behind it especially due to reduction of the xylem vessel structures under drought. Author should specially focus that “drought reduced hydraulic failure of the plant under the drought stress condition and reduction the plant water status by reducing the leaf water potential and sap movement due to reduction (no. diameter, area) of the xylem vessel structure” Mention this point in your paragraph of 360-370 and cite the above mention article as a reference.

10) Conclusion section (Line no. 403)

  1. Author should have revised the conclusion little more because its seems like direct repetition of the result part, I agree that the result should be include some part but tone of the presentation should be different. Please correct the 405-412. Conclusion section should be in good in flow with include the all-necessary components and it should focus the future insight of the research based on your current findings.

11) Reference (Line no. 436): please double-check the citations, their style, and spell check, and other grammatical errors. moreover, I request to authors for revision throughout the manuscript according to the journal rules.

Good Luck!

Author Response

General comments

I have read the manuscript (forests -1481157). Entitle: Drought drives growth and mortality rates in three pine species under Mediterranean conditions written by Cristina Valeriano et. al., for publication of forests MDPI. In this study, the author investigates the Mortality and growth rate of three pine species in could of years’ research (2012-2017). Author found that the death tree found that slow growth. Author also found that the when climate is shifted to drier and warmer the growth response of the tree more positively to wet climates conditions in living than the dead trees. Author also found that the mortality is increased if acidification continues.

The overall research is well conducted and research is obvious application potential for the readers because this study reflected the cumulative drought impact on tree functional mortality. In this sense, the manuscript is much valuable. However, I found some points especially the flow of the text is not smooth and sometimes I found the shallow writing and lack of potential references, and lack of connection of story in different paragraphs especially in the introduction and discussion sections. In discussion, the author should be deal with the physiological perspectives and what will affect plant biology. I also found the lack of potential and appropriate references to support the findings. The author should provide enough examples and their interpretation of different traits of physiological and biochemical responses. I mention some tips and recommended some literature this will help to improve this manuscript quality better than before. Overall after I evaluate this manuscript, I request the author for the “MAJOR REVISION” and also, I request to authors for revision according to the rules of the journal and correct the bibliography.

We thank you for your revision and comments.

 Major suggestions

1) Abstract Issue: The author wrote the important finding in their abstract but the text seems like comparatively more description. Please concise the text little bit more with only focus the major outcomes. Abstract should more logical, short, concise, and informative and it should reflect your study and major findings while shortly observed by readers. Please make the necessary corrections. Also please follow the pattern and author pack and previously issued articles, I think the separate “section title” is not needed just author may write thoroughly without separation of subtitle in the abstracts.

Response 1.- Thank you for recognition our findings, we eliminated the section titles.

2) Introduction: I satisfy with the introduction except the starting parts. Please improve your first and second paragraph more logically. Author should to be integrate the drought component and negative impact in plant biology, however author directly jump the part of rising temperature, water scarcity and then tree mortality. For the sequential flow and for the enough background author should deal the negative consequences of drought stress in this climate change era. The negative effect of drought for the plant biological points of should present with the enough reference before directly go to the tree death part. Please read and cite article entitle: “Response of drought stress in prunus sargentii and larix kaempferii ...https://doi.org/10.1016/j.foreco.2020.118099” and mention that “drought reduced the morphological and physiological traits, reduce the leaf water potential and sap movement due to alternation of xylem anatomical features in the plants”. Then only author should go to the area description and water scarcity scenario (ie. Southern Europe) and thoroughly make the background and raise questions and hypothesis of the study about the forest mortality due to cause of drought.

Response 2.- We improved the flow among paragraphs as you suggested for the Introduction.

We appreciate the recommendation paper, but we preferred not to add this citation because neither the species, type of forest nor the site (Mediterranean ecosystems) are alike to be used as a comparison with our study.

 3) Hypothesis of the study: Author well describe the aim of the study from the line no 88-93, which are more satisfactory. However, author should include the section of the hypothesis of the study as well in the last section of the introduction. Based on the hypothesis author should mentioned the aim of the study. In past paragraph of starting author raise the problem of specific area of Spain and tree mortality and dieback problem due to drought stress which is good text but author also should have mentioned hypothesis of the study for e.g. We hypothesized that these three species are contrasting pattern of physiological and morphological response after exposed to the drought stress therefore…. something like this way. Hypothesis of the study is important thing and it give another strength for the introduction. The hypothesis should be very clear in the introduction sections because, without appropriate literature, questions, or hypotheses in the introduction section the entire text will be unclear. The author should give special attention and the sequential presentation this section in the last part of introduction.

Response 3.- We included a clearly stated hypothesis before the aim.

 Some others suggestions

4) Table 2 (Line no. 131): Author should full mention their abbreviation if those are used inside the in the footnote, see DBH it probably may be “diameter breast height”. Each table are independent with full information.

Response 4.- We have included the abbreviations on the table caption.

5) Materials and methods: Author should be clearly mentioning how author recognize the tree drought severity and its mortality author mentioned somewhat unclear. Please revised this clearly method in MM section because this is main part for your study in different site to identify the drought stress.

Response 5.- We included a sentence to clarify this issue and how we recognized drought-killed trees.

6) Figure1, 2, 3…: Please little more increased the font of the letter inside the figures, it seems like difficult to recognized. Also please increased in x-axis and Y axis as well and make this consistence of all the figures.

Response 6.- We corrected the size of the labels of these figures.

7) Line no. 322: It is somewhat unclear and also have opposite meaning “living tree showed higher growth rate than dead trees” what you this means. It is always true but I am expecting advance text rather than this. Please revised this.

Response 7.- This sentence is a description of the results in table 3 showing a higher growth in living than dead trees with the exception of Orera site.

8) Line no. 328: Author mention that the study region is affected due to drought in the study region (North east Spain) since 2010 and give the citation. That means this type of tree mortality region the study was already conducted before you, if you should have showed the addition thing and should your report differ than that literature.  Please consider this point more seriously.  

Response 8.- Our study is novel; the reference to Vicente-Serrano et al. (2021) study is because they analyzed long-term variability and trends in droughts using the Standardized Precipitation Index (SPEI). They just focused on drought occurrence and climate data, but not on the impacts on forests.

 9) Line no. 360-372: Whole of this paragraph in the discussion should be clearer, especially author should focus the line from 363-370. I agree with the author statement that the tree mortality is directly related due to formation of wider conducts that cause make embolism and loss of hydraulic conductivity. In here author should describe more this statement especially anatomical (xylem vessel structure) point of view. Please read this article: Impact of drought stress on photosynthesis responses, leaf water and sap flow…... “DOI:10.1016/j.scienta.2018.11.021 This article describe very well about the hydraulic conductance of the plant and reduction of hydraulic failure and its possible reason behind it especially due to reduction of the xylem vessel structures under drought. Author should specially focus that “drought reduced hydraulic failure of the plant under the drought stress condition and reduction the plant water status by reducing the leaf water potential and sap movement due to reduction (no. diameter, area) of the xylem vessel structure” Mention this point in your paragraph of 360-370 and cite the above mention article as a reference.

Response 9.- We have added your recommended reference and some additional studies related to our study site in Mediterranean ecosystems for improving the discussion. Thanks.

10) Conclusion section (Line no. 403)

Author should have revised the conclusion little more because it seems like direct repetition of the result part, I agree that the result should be include some part but tone of the presentation should be different. Please correct the 405-412. Conclusion section should be in good in flow with include the all-necessary components and it should focus the future insight of the research based on your current findings.

Response 10.- We revised and corrected the conclusion to improve the flow.

11) Reference (Line no. 436): please double-check the citations, their style, and spell check, and other grammatical errors. moreover, I request to authors for revision throughout the manuscript according to the journal rules.

Response 11.- We corrected all the formatting errors from the references.

Good Luck!

Thank you.

Reviewer 2 Report

General remarks of the reviewer

Title: The title of the article is accurate and directly relates to the purpose of the research.

Abstract: The abstract gives a good overview of the work.

Keywords: The keywords are specific to the topic under study.

Introduction: The state of the research  is reviewed  and major publications cited.

Materials and Methods:

2.2. Field Sampling

Please explain the directions from which the two cores were taken at  height of 1.3 m.

Results:

In M&M (2.2. Field Sampling) there is information about the assessment of crown defoliation of all examined trees, but there is no interpretation of this feature in the results.

3.2. Growth patterns

In order to objectively compare basal area increment and tree-ring width, it is necessary to know the dbh without bark mean for ND and D (in the arrangement as in Table 2 - dbh averages), because they depend primarily (mathematically) on the dbh size.

Discussion: The research results are well-discussed.

Conclusions: The conclusions are constructive.

References: The subject literature is sufficient. Please correct authors and book publications for items: 3-5, 9, 13-14, 17, 19, 21, 24, 27, 32-33, 36-37, 50, 56. 60, 62, 65, 69, 71 and 75.

Technical Notes

The description of the literature item needs to be corrected as required by the publisher: articles, books and other sources - italics of journal titles, year in bold, correct pages of journals and the access link and date of access in English. According to MDPI standard.

Details in the attached manuscript.

Summary of the review:

The article exhausts the presented issue.

Author Response

General remarks of the reviewer

Title: The title of the article is accurate and directly relates to the purpose of the research.

Abstract: The abstract gives a good overview of the work.

Keywords: The keywords are specific to the topic under study.

Introduction: The state of the research is reviewed and major publications cited.

Thank you. We appreciate the positive feedback.

Materials and Methods:

2.2. Field Sampling

Please explain the directions from which the two cores were taken at height of 1.3 m.

We took the cores parallel to slope, to avoid sampling reaction wood.

Results:

In M&M (2.2. Field Sampling) there is information about the assessment of crown defoliation of all examined trees, but there is no interpretation of this feature in the results.

3.2. Growth patterns

In order to objectively compare basal area increment and tree-ring width, it is necessary to know the dbh without bark mean for ND and D (in the arrangement as in Table 2 - dbh averages), because they depend primarily (mathematically) on the dbh size.

We have included a sentence according to the reviewer’s suggestion.

Discussion: The research results are well-discussed.

Conclusions: The conclusions are constructive.

Thank you.

References: The subject literature is sufficient. Please correct authors and book publications for items: 3-5, 9, 13-14, 17, 19, 21, 24, 27, 32-33, 36-37, 50, 56. 60, 62, 65, 69, 71 and 75.

Technical Notes

The description of the literature item needs to be corrected as required by the publisher: articles, books and other sources - italics of journal titles, year in bold, correct pages of journals and the access link and date of access in English. According to MDPI standard.

We corrected all the formatting errors from the references.

Details in the attached manuscript.

Summary of the review:

The article exhausts the presented issue.

Thank you for remarked the errors in the paper, it helps!

Reviewer 3 Report

Dear authors

(forests-1481157)

Drought drives growth and mortality rates in three pine species under Mediterranean conditions

In the current investigation, the authors studied the cumulative effects of climate change especially drought stress with warming on three pine species (P. sylvestris, P. pinaster and P. halepensis) grown in north-eastern Spain. They reconstructed death dates and growth rates of three studied species based on climate data which were determined during the period from 1970-2020. Eight sites were selected to observe the recent signs of extensive canopy dieback and elevated mortality rates linked to recent droughts. Furthermore, some variables were determined such as basal area, density, mortality percentage, diameter at breast height (Dbh), number of sampled dead / living trees according to canopy defoliation (%).

Several statistical analyses were done to get the link between growth trends and the response to drought of living and dead trees

This kind of research has a significant importance at the academic and economic levels; it can give us a comprehensive picture to identify the vulnerable forests or tree populations that may show extinction processes with the frequent climate change in the Mediterranean basin region.

Generally, the overall quality of this manuscript is good and written well. However, I think some issues should be addressed before considering this manuscript satisfactory for publishing

Here, these points can be summarized:

  • In table 1, 2 and 3, there is on obvious overlap between the three studied species and the different sites. Please separate them by spaces or lines to be easier to the reader
  • In table 1, Climate data correspond to the 1970-2020 period. It is too long period. Are the data in this table the means of 50 years? It is not logic especially with the long period and the frequent climate change
  • In table 2, Dbh is an abbreviation without explanation. Please add Diameter at breast height
  • Line 170, 1.1-km2. Please correct the superscript.
  • The same issue in line 174, 180, 181, 184. Please check these issues of superscripts throughout the manuscript
  • Where are the letters of the statistical analysis in table 3?
  • In this study, I don’t know why authors didn’t try to study some physiological aspects i.e. photosynthesis, the rate of carbon fixation, water use efficiency, transpiration, hydraulic features, stomatal conductance… etc. These variables may give us good comparison between different species with climate change in different sites. Also, we can see from these features an early-warning signal to the effect of drought on the distribution of tree populations  
  • Furthermore, why authors did not analyze wood density by X-ray or different wood anatomical variables. Climate change can affect disease outbreaks leading to reduce growth and mortality
  • The authors focused on the effect of drought stress even in the title of manuscript, it is not true, climate change consist of different factors such as heat stress besides drought can cause mortality and extensive dieback. I would like to know the response of authors on this point.
  • The authors mentioned that their findings support forecasts of higher vulnerability of some tree populations in the Mediterranean basin in response to a warmer and drier climate. How can we use these data to keep the biodiversity and ecosystems in this important area of the world? Please add this point in the conclusion and to the objective of this study.

Thank you and all the best wishes

Author Response

Dear authors

(forests-1481157) Drought drives growth and mortality rates in three pine species under Mediterranean conditions

In the current investigation, the authors studied the cumulative effects of climate change especially drought stress with warming on three pine species (P. sylvestris, P. pinaster and P. halepensis) grown in north-eastern Spain. They reconstructed death dates and growth rates of three studied species based on climate data which were determined during the period from 1970-2020. Eight sites were selected to observe the recent signs of extensive canopy dieback and elevated mortality rates linked to recent droughts. Furthermore, some variables were determined such as basal area, density, mortality percentage, diameter at breast height (Dbh), number of sampled dead / living trees according to canopy defoliation (%).

Several statistical analyses were done to get the link between growth trends and the response to drought of living and dead trees

This kind of research has a significant importance at the academic and economic levels; it can give us a comprehensive picture to identify the vulnerable forests or tree populations that may show extinction processes with the frequent climate change in the Mediterranean basin region.

Generally, the overall quality of this manuscript is good and written well. However, I think some issues should be addressed before considering this manuscript satisfactory for publishing

We thank the reviewer for his/her constructive comments. And we try to fix the issues to improve it.

Here, these points can be summarized:

  • In table 1, 2 and 3, there is on obvious overlap between the three studied species and the different sites. Please separate them by spaces or lines to be easier to the reader

We put some lines to separate better the species and sites.

  • In table 1, Climate data correspond to the 1970-2020 period. It is too long period. Are the data in this table the means of 50 years? It is not logic especially with the long period and the frequent climate change

We used the same period in all de climate analyses and we observed that since 1979 the frequency of droughts and their intensity have increased due to climate change. It can be seen in figure 1B.

  • In table 2, Dbh is an abbreviation without explanation. Please add Diameter at breast height

We have included the abbreviation on the table caption.

  • Line 170, 1.1-km2. Please correct the superscript.

We have corrected the superscript.

  • The same issue in line 174, 180, 181, 184. Please check these issues of superscripts throughout the manuscript

We have corrected these typos too.

  • Where are the letters of the statistical analysis in table 3?

There are letters only in Corbalán site because is the only one with significant differences.

  • In this study, I don’t know why authors didn’t try to study some physiological aspects i.e. photosynthesis, the rate of carbon fixation, water use efficiency, transpiration, hydraulic features, stomatal conductance… etc. These variables may give us good comparison between different species with climate change in different sites. Also, we can see from these features an early-warning signal to the effect of drought on the distribution of tree populations  

Thank you for the recommendation. We will consider it for future studies but in this study we were only interested on using radial growth data and mortality dates as indicators of drought stress and impacts on forests.

  • Furthermore, why authors did not analyze wood density by X-ray or different wood anatomical variables. Climate change can affect disease outbreaks leading to reduce growth and mortality

We are in process to analyzing wood anatomy in some sites, but we have not observed diseases or pest attacked in the affected stands. We avoided sampling forests affected by those biotic stress factors.

  • The authors focused on the effect of drought stress even in the title of manuscript, it is not true, climate change consist of different factors such as heat stress besides drought can cause mortality and extensive dieback. I would like to know the response of authors on this point.

              There are many factors that can affect trees causing death and dieback; we agree. However, we focused on the impacts of a particularly climate extreme, drought, which is difficult to disentangle of climate trends such as the rise in temperature and VPD. However, the facts that the peaks in mortality and growth drops followed recent severe drought make us confident on our results and on our title.

  • The authors mentioned that their findings support forecasts of higher vulnerability of some tree populations in the Mediterranean basin in response to a warmer and drier climate. How can we use these data to keep the biodiversity and ecosystems in this important area of the world? Please add this point in the conclusion and to the objective of this study.

We can use this data to identify “dieback hotpots” where non-linear responses to drought involve high damage and mortality rates. These stands could be used as sentinels for monitoring the responses of vulnerable forest ecosystems to climate warming and aridification. This would allow identifying the most vulnerable populations, species of forests. We added similar comment to the Conclusions.

Thank you and all the best wishes

Thanks a lot for your constructive review.

Round 2

Reviewer 1 Report

Dear Author

I have read the manuscript (forests -1481157). Entitle: Drought drives growth and mortality rates in three pine species under Mediterranean conditions written by Cristina Valeriano et. al., for publication of forests MDPI.

Author addressed all the questions and comments what I suggested in the original manuscript. I satisfy the author revisions throughout the paper. The abstract issue is well written by the author. Now this manuscript improved the flow of the flow of writing, which was comparatively shallow in the original version but in this revised copy author address all the quarries and suggestions where the introduction is significantly improved by author. Before accept this manuscript, I request to author to check the whole manuscript by native speaker for correct spell check, and other grammatical errors.

Author Response

Thank you.

We carefully checked the English usage and corrected grammatical erros with the assistance of native speakers.

Reviewer 3 Report

Dear authors

Thank you for your responses

Most of issues have been addressed in the revised version of your manuscript 

All the best wishes

Author Response

Thank you.